# A Scoping Review on the Prevalence of Hashimoto’s Thyroiditis and the Possible Associated Factors

**DOI:** 10.3390/medsci13020043

**Published:** 2025-04-10

**Authors:** Hernando Vargas-Uricoechea, Alejandro Castellanos-Pinedo, Karen Urrego-Noguera, María V. Pinzón-Fernández, Ivonne A. Meza-Cabrera, Hernando Vargas-Sierra

**Affiliations:** 1Metabolic Diseases Study Group, Department of Internal Medicine, Universidad del Cauca, Carrera 6 Nº 13N-50, Popayán 190001, Colombia; karenurrego@unicauca.edu.co (K.U.-N.); mpinzon@unicauca.edu.co (M.V.P.-F.); imeza@unicauca.edu.co (I.A.M.-C.); hdvargas@unicauca.edu.co (H.V.-S.); 2Faculty of Medicine, Universidad del Sinú, Hospital San Jerónimo, Montería 230001, Colombia; acaspinedo@yahoo.es; 3Health Research Group, Department of Internal Medicine, Universidad del Cauca, Popayán 190003, Colombia

**Keywords:** Hashimoto, thyroiditis, prevalence, autoimmunity, thyroid

## Abstract

Background: Hashimoto’s thyroiditis (HT) is the most common autoimmune thyroid disease (AITD) and is characterized by the presence of thyroid autoantibodies against thyroid peroxidase and/or thyroglobulin. Several studies have found that the global prevalence of HT has increased in recent decades, while others show the opposite. Methods and Results: The objective of this scoping review was to synthesize and analyze the different studies that have evaluated the prevalence of HT (in adults) and the possible associated factors. The following databases were consulted, as follows: MEDLINE, Web of Science, PubMed, and Scopus. The search terms “epidemiology”, “prevalence”, and “Hashimoto disease” and “Hashimoto thyroiditis” were used. The search was limited to articles published between January 1965 and October 2024, and only articles in English were considered. In order to reduce selection bias, each article was scrutinized using the JBI Critical Appraisal Checklist independently by two authors. Studies were included if the number of participants (study population and/or cases and controls, depending on the study design) was clearly described and duplicate studies were excluded. A total of 59 studies were identified, the vast majority of them used a cross-sectional design, using different methods of disease assessment. Conclusions: Globally, the prevalence of HT is estimated to be between 5–10%; some areas with prevalences > 20% and others < 0.5% were identified. Prevalence is also higher in women than in men. Multiple underlying factors (genetic, epigenetic, environmental, and lifestyle), together with socioeconomic, nutritional, overdiagnosis, inter alia, may explain (at least in part) the wide variability in the prevalence of HT.

## 1. Introduction

Although there is no consensus on the definitions of autoimmunity and autoimmune diseases (AIDs), it is generally accepted that autoimmunity is a response involving multiple autoreactive adaptive immune components, whereas an AID involves one or more clinically evident pathologies [1].

AIDs are caused by a combination of multiple genetic, epigenetic, environmental, and lifestyle factors. The most prominent immunological manifestation of AIDs is the production of autoantibodies (Abs), which are the most valuable biomarkers for diagnosis (and even for disease classification and activity) [1,2,3,4].

Classically, it has been established that AIDs can be restricted according to the pattern of organic involvement (e.g., organ-specific) or be generalized or systemic (non-organ-specific). In this sense, autoimmune thyroid diseases (AITDs) are organ-specific AIDs [3,4,5].

AITDs are a group of diseases characterized by the breakdown of tolerance to thyroid autoantigens (specifically, thyroid peroxidase—TPO; thyroglobulin—Tg; and the thyrotropin receptor—TR), which induce both humoral and cellular autoimmune responses, with the presence of circulating Abs to thyroid autoantigens (TgAbs, TPOAbs, and TRAbs) and lymphocytic infiltration [6,7].

The clinical spectrum of AITDs includes Hashimoto’s thyroiditis (HT), Graves–Basedow disease, postpartum thyroiditis, drug-induced thyroiditis, thyroiditis associated with polyglandular syndromes, and atrophic thyroiditis [6,7,8]. Among all AITDs, HT is the most common, and it often manifests as hypothyroidism (subclinical or overt) as a consequence of thyrocyte destruction (a phenomenon mediated by both humoral and cellular immunity) [6,9,10].

HT is the main cause of hypothyroidism in iodine-sufficient areas of the world [11]. Its global prevalence varies depending on various factors (e.g., socioeconomic, environmental, and genetic) and is estimated at 4.8–25.8% and 0.9–7.9% in men and women, respectively [11,12,13]. However, some studies have reported an increase in the prevalence of HT in recent decades, while other studies have shown a slight decrease [11,12,13].

The objectives of this study are to determine the global prevalence of HT and analyze the possible factors that may influence the population variability of the disease.

## 2. Materials and Methods

This scoping review is part of a larger research project investigating and evaluating the distribution and behavior of AITDs in Colombia. The protocol was developed in accordance with the principles of the Declaration of Helsinki and approved by the Research Ethics Committee of the University of Cauca, Colombia (ID: 4656, January 2018).

### 2.1. Literature Search and Selection Criteria

A comprehensive and sensitive search was performed in the MEDLINE, Web of Science, PubMed, and Scopus databases. The search was limited to articles published between January 1965 and October 2024 (adults, humans, clinical trials, meta-analyses, reviews, and systematic reviews), and only articles in English were considered.

The search criteria were as follows: Epidemiology [Title/Abstract] OR Prevalence [Title/Abstract] AND Hashimoto Disease [Mesh] OR Hashimoto Thyroiditis [Title/Abstract] OR Chronic Lymphocytic Thyroiditis [Title/Abstract] OR Lymphocytic Thyroiditis, Chronic [Title/Abstract] OR Thyroiditis, Chronic Lymphocytic [Title/Abstract] OR Hashimoto’s Disease [Title/Abstract] OR Disease, and Hashimoto’s [Title/Abstract] OR Autoimmune thyroiditis [Title/Abstract].

Studies were included if the number of participants (study population and/or cases and controls, depending on the study design) was clearly described, and duplicate studies were excluded (Figure 1).

### 2.2. Data Extraction

To reduce selection bias, each article was scrutinized according to the JBI Critical Appraisal Checklist [14]. The identification and selection of studies were carried out by two authors (H V–U and H V–S). Data were extracted using a standardized template using a predefined data form created in Excel, and in cases where discrepancies arose in the extracted data, they collaboratively conducted a second round of extraction to validate the accuracy of the information.

### 2.3. Data Analysis

The following data were retrieved as follows: authors, year of publication, country (income group), number of cases and population evaluated, study design, diagnostic criteria, and prevalence of HT (95% CI). No statistical analysis or meta-analysis was performed due to the high heterogeneity observed among the studies included in this review. However, we conducted a descriptive analysis to summarize key study characteristics. This scoping review was registered on the “International Platform of Registered Systematic Review and Meta-Analysis Protocols INPLASY” (registration number: INPLASY202530049; DOI number: 10.37766/inplasy2025.3.0049) and followed the “Reporting Checklist for Systematic Review. Based on the PRISMA guidelines”.

## 3. Results

### 3.1. Origins of Studies and Diagnostic Criteria Used

A total of 59 studies were identified (25 in Europe, 5 in Africa and Oceania, 19 in Asia, 4 in North America, and 6 in South America) [15,16,17,18,19,20,21,22,23,24,25,26,27,28,29,30,31,32,33,34,35,36,37,38,39,40,41,42,43,44,45,46,47,48,49,50,51,52,53,54,55,56,57,58,59,60,61,62,63,64,65,66,67,68,69,70,71,72,73]. Fifty studies used a cross-sectional design, while two were cohort studies, and seven were array research. The methods used for HT diagnosis were as follows: serum Abs; thyroid ultrasound (TU); thyroid tissue; serum Abs + TU; serum Abs + TU + fine–needle aspiration (FNA); and thyroid tissue + Abs + TU. In four studies, the diagnostic method used was not reported.

The 59 identified studies were conducted in 25 countries: 13 were high income (HI) (England, Finland, Norway, Germany, Italy, Denmark, Spain, Poland, Croatia, Australia, Japan, South Korea, and USA); 7 were upper-middle income (UMI) (Brazil, China, Russia, Bosnia and Herzegovina, Jordan, Mexico, and Colombia); and 5 were lower-middle income (LMI) (Tunisia, Nigeria, Ghana, Iran, and Sri Lanka) [15,16,17,18,19,20,21,22,23,24,25,26,27,28,29,30,31,32,33,34,35,36,37,38,39,40,41,42,43,44,45,46,47,48,49,50,51,52,53,54,55,56,57,58,59,60,61,62,63,64,65,66,67,68,69,70,71,72,73].

### 3.2. Global Prevalence of HT

Globally, the prevalence of HT is between 5% and 10%; however, prevalence rates were widely variable between continents, between countries, and even between the regions studied (within countries) [10,11,12,13]. The prevalence is also higher in women than in men (between 2.0 and 7.0 times higher) [10,11,13].

### 3.3. Prevalence of HT in Europe

In Europe, prevalence rates as low as 0.42% and 0.43% were found in Russia and in Bosnia and Herzegovina, respectively, and very high prevalence rates were found in Italy (35.5%) and Denmark (39.7%), while other studies conducted in these same countries found lower prevalence rates [15,16,17,18,19,20,21,22,23,24,25,26,27,28,29,30,31,32,33,34,35,36,37,38,39] (Table 1).

### 3.4. Prevalence of HT in Africa

Similarly, the prevalence in Africa was also variable, with the highest prevalence in Tunisia (22.8%) and the lowest in Nigeria (6.7%) [40,41,42] (Table 2).

### 3.5. Prevalence of HT in Oceania

The studies conducted in Oceania (Australia) reported prevalences of 13.1% and 8.6% [43,44] (Table 3).

### 3.6. Prevalence of HT in Asia

In Asia, prevalence rates as low as 0.1% in South Korea and 0.3% in China were found; however, in China, the reported prevalence rates differed substantially, depending on the area studied. The highest prevalence rates were documented in Japan (18%), China (16.1%), and Jordan (15.1%) [45,46,47,48,49,50,51,52,53,54,55,56,57,58,59,60,61,62,63] (Table 4).

### 3.7. Prevalence of HT in North America and South America

Finally, in North America and South America, the lowest prevalences were found in the USA (0.4%) and Brazil (0.1%), respectively, although some studies carried out in the USA showed very high prevalences. The highest prevalences documented were in the USA (22.4%) and in Colombia (22.3%) [64,65,66,67,68,69,70,71,72,73] (Table 5 and Table 6).

## 4. Discussion

The prevalence of HT worldwide is widely variable between different continents, between different countries, and even within the same country.

The variability found in the prevalence of HT has also been demonstrated in multiple AIDs and could be explained by a constellation of factors [14], among which the following stand out:

### 4.1. Sex and Age

A characteristic of many AIDs is that they are more frequent in women and become more common as the age of the population increases. A similar phenomenon occurs in HT: multiple studies have shown that the diagnosis of hypothyroidism (due to HT) is more frequent in women than in men, and the incidence of diagnosis increases with age (with a high proportion of TPOAb and/or TgAb positivity). In fact, age and the presence of thyroid Abs have been identified as predictive factors for hypothyroidism in older people [39,74,75,76,77,78,79].

### 4.2. Socioeconomic Status and Disparities in Health Care and Ethnicity

In general terms, the most disadvantaged socioeconomic groups have a higher frequency of AIDs, which suggests a socioeconomic gradient that may explain the variability in these diseases (including HT). In addition, in populations where there are disparities in health care and in those where there are racial and ethnic minorities, differences are found not only in the diagnostic procedures of different diseases but also in the results of treatment and prognosis [80,81,82,83,84]. It is thus possible that the population prevalence of HT may be affected by large socioeconomic disparities (or the presence of ethnic minorities).

### 4.3. Seasonality

An area of uncertainty has been the possible association between the month of birth and the risk of HT. Some studies have found significant differences in the month of birth between individuals with HT and controls; that is, individuals with HT were more likely to be born in summer (June–August), supporting the hypothesis that seasonal variations may contribute to the development of HT. Other studies have analyzed the seasonal pattern of the birth months of young patients with HT, finding significant differences among those with HT; e.g., in a month-by-month analysis, the highest and lowest predispositions (for HT) were among those born in spring (March) and autumn (November), respectively. Season type (at birth) and sex have also been found to be factors related to HT, even after adjustment for other factors, such as the year of birth and age [85,86,87,88].

### 4.4. Racial Variations and Genetic Polymorphisms

Several susceptibility genes have been implicated in the risk of developing AITDs, e.g., genes specific to the thyroid or those that interfere with immunomodulation phenomena. Single-nucleotide polymorphisms (SNPs) in these genes may affect both the regulation of central or peripheral tolerance and the processes of T-cell activation and antigen presentation, inducing an AITD. Therefore, these SNPs may explain the phenotypic diversity among individuals affected by HT. Some studies have suggested that racial variations in the prevalence of AITDs accentuate possible SNPs and increase genetic susceptibility to HT (although it is widely accepted that combinations of different SNPs can increase the risk of AITDs, they can also have the opposite effect) [89,90,91,92].

### 4.5. The Kaleidoscope of Autoimmunity

The kaleidoscope of autoimmunity indicates the possible variations in the clinical spectrum of an AID (or the probability that an AID may co-occur in the same individual (polyautoimmunity) or that several AIDs coexist in the same family (familial autoimmunity)). In this sense, HT is one of the AIDs that are likely to co-occur with other AIDs (e.g., rheumatoid arthritis, diabetes mellitus type 1, and systemic lupus erythematosus). The risk of co-occurring AIDs is greater than expected by chance, and the presence of an AID can be a predictor of a second AID; thus, in the study of any AID, the AID itself increases the likelihood of identifying one or more other AIDs, which can affect the prevalence of other diseases with a similar autoimmune basis (e.g., HT) [93,94,95,96].

### 4.6. Obesity

Obesity may exacerbate the progression and severity of several AIDs. Although the mechanisms linking BMI to HT are not entirely clear, some studies suggest a possible association between a high BMI and HT [97,98,99]. Thus, considering that global adult obesity has more than doubled since 1990 and that, in 2022, 2.5 billion adults were overweight (and of these, 890 million were obese), it can be argued that changes in HT prevalence may be somewhat influenced by the overweight/obesity pandemic [100,101,102,103].

### 4.7. Environmental Factors and Contaminants

It is widely accepted that a number of environmental factors, including climate change, pollution, nutritional changes, and thyroidal endocrine-disrupting chemicals, affect thyroid function [104,105]. Some studies have found associations between environmental factors, contaminants, endocrine disruptors, and AITDs. Therefore, the variability in the global prevalence of HT may be influenced by these factors [106,107,108,109,110].

### 4.8. Smoking and Alcohol Intake

Traditionally, smoking has been considered to affect thyroid function and to be capable of inducing AITDs, as well as other AIDs; however, such effects are controversial. To date, it has been difficult to establish a link between smoking and HT, especially because this influence is exerted through multiple mechanisms that can be modified by several factors (age, sex, ethnicity, and iodine status) [111,112,113,114,115]. Current evidence suggests that smoking is associated with a decrease in TSH levels and an increase in TPOAb levels (especially in men) [114,116,117,118]. For this reason, the prevalence of HT could be influenced by the frequency of smoking in the individuals evaluated.

It has also been suggested that alcohol consumption reduces T4 levels and that T4 and T3 levels decrease in alcoholic individuals after alcohol abstinence. Since smoking frequently coincides with alcohol intake, it is necessary to take alcohol consumption into account when evaluating the association between smoking and AITDs [114,119,120,121]. However, studies evaluating the effect of smoking on thyroid function (or thyroid autoimmunity) have not determined whether such associations are independent of alcohol consumption and/or other factors.

### 4.9. Iodine Intake and Population Iodine Status

Disorders associated with iodine intake (especially iodine deficiency disorders (IDDs)) are highly frequent worldwide; universal salt iodization (UIS) programs were designed as a cost-effective strategy for the prevention of IDDs [122]. However, these programs are not always carried out permanently or systematically, which explains why there are still countries with iodine deficiency or excess in the population. This has led to an increase in the frequency of functional and immunological thyroid disorders (and probably thyroid cancer) [123,124,125]. Some studies have shown an increase in the prevalence of thyroid TPOAbs and/or TgAbs during the first years of implementation of UIS programs, while others have found a decrease in prevalence when UIS programs are sustained over time. These findings are mainly described in areas with prolonged iodine deficiency (and that implemented USI programs) [126]. Therefore, population iodine status should be considered a factor influencing the prevalence of HT.

### 4.10. The Population Status of Other Micronutrients

Several studies have found a possible association between deficiencies of some micronutrients (selenium, iron, copper, magnesium, vitamin B12, vitamin D, etc.) and a higher frequency of TPOAb and TgAb positivity; in fact, the risk increases as the micronutrient deficiency becomes more severe [127,128,129,130,131,132]. Despite this, intervention studies have been contradictory, as correction or supplementation with the deficient micronutrient does not always translate into a decrease in the risk of AITDs or in the level of thyroid Abs [133,134,135,136,137,138,139]. Based on the above and the fact that micronutrient deficiencies affect a large proportion of the world population, such deficiencies may affect the prevalence of HT [140,141,142,143,144], especially since famines and undernutrition/malnutrition remain unacceptably high; e.g., around 733 million people in the world were found to be undernourished in 2023 (9% of the world population). This represents an increase of 36% compared to 2014 (almost 539 million people affected) [145]. Hence, the nutritional component of the world population is associated with the population status of micronutrients, some of which have been shown to be associated with HT.

### 4.11. Overdiagnosis

The frequency of hypothyroidism increases with age, and its main causes globally are iodine deficiency and HT [146,147]. Therefore, the high prevalence of IDDs and thyroid Ab positivity clearly sets a precedent when analyzing the global prevalence of HT; however, as the symptoms of hypothyroidism may overlap with those of other chronic diseases, it is more likely that thyroid function tests (and measurements of thyroid Abs) are performed indiscriminately in individuals with such symptoms, identifying the disease (HT) earlier; it may even lead to overtreatment (by giving treatment with LT4) in individuals with subclinical hypothyroidism when it is probably not required [148,149,150,151]. This also extends to women who wish to become pregnant (or those already pregnant), where overscreening frequently occurs in the search for autoimmunity and/or thyroid functional disorders, which may modify the overall prevalence of HT [13,152,153,154].

### 4.12. Temporal Trends

The frequency of the diagnosis of hypothyroidism (and HT) has changed over time and varies according to the geographical area studied. While some studies show a decrease in frequency, others report an increase [155,156,157]. The increase in frequency can be explained by the increase in the diagnosis of most AIDs (due to a real increase in frequency, the greater availability of laboratory tests focused on the diagnosis of autoimmunity, or overscreening/overdiagnosis, especially in the approach to hypothyroidism/HT) [158,159,160,161]. Meanwhile, the decrease in frequency could be explained (at least in part) by different international guidelines and recommendations (in the last two decades) aimed at preventing the overdiagnosis of hypothyroidism by establishing a diagnostic approach based on the TSH value, which was previously not carried out routinely before starting treatment with LT4. These recommendations were made with the aim of reducing LT4 overtreatment, and the effect took several years to become evident [162,163,164,165]. For example, a study evaluating the initiation of new LT4 prescriptions (between 2006 and 2012) found that about 30% of these treatments were not based on a previous TSH measurement; additionally, these patients were started on LT4 replacement therapy according to the symptoms that they manifested (such as weight gain, constipation, fatigue, muscle weakness, or hair loss) [166]. This overdiagnosis/overtreatment may (falsely) increase the prevalence of hypothyroidism/HT; in fact, a greater overtreatment has been documented in older adults, which may also affect the prevalence in younger individuals. Based on this aspect, international recommendations require a reasonable time to be established in the international medical community (and may explain the temporal trend showing a decrease in the prevalence of HT in several countries) [166].

### 4.13. Temporal Trends in Thyroid Cancer

The incidence of thyroid cancer (TC) is increasing annually worldwide, with a growing burden. Several reasons for the increase in the frequency of TC have been put forward, including overdiagnosis and a greater access to imaging techniques [167,168,169]. In general terms, the evaluation of patients with a suspected or confirmed diagnosis of TC requires the assessment of thyroid function (and, in a significant proportion of them, the value of Tg and thyroid Abs), which may increase the probability of identifying an AITD earlier (with or without thyroid functional alterations) and may eventually affect the prevalence of HT.

### 4.14. Other Factors

Other factors may affect the prevalence of HT. For example, COVID-19/SARS-CoV-2 vaccines have been associated with an increase in both TPOAb and TgAb levels, although the findings have been inconsistent across studies. Similarly, other viral infections have also been associated (Epstein–Barr, parvovirus B19, herpesvirus 6A, enterovirus, chronic HBV and HCV infections, etc.) with HT [170,171,172,173,174]. Meanwhile, some nutritional components, especially with the rise of “fad diets” (specifically those free of gluten and/or lactose), have been suggested to affect the natural course and prevalence of HT [174]. Therefore, infectious (viral) and nutritional components may be affecting the prevalence of HT in different clinical settings.

### 4.15. Diagnostic Criteria and the Definition of the Disease

The diagnostic criteria for HT have not been universally accepted in the studies evaluated in this review. This aspect affects the overall prevalence of the disease, since the results expressed as diagnoses based on only the levels of TPOAbs and TgAbs are different from those in studies with diagnostic criteria based on thyroid tissue samples or TU (or with a combination of evaluation methods) [175].

### 4.16. Drugs

Some medications may be associated with an increased risk of AITDs or thyroid function disorders; e.g., the increasingly widespread use (in various clinical settings) of medications such as amiodarone, tyrosine kinase inhibitors, immune checkpoint inhibitors therapy, PD-(L)1-Ab, lithium, interferon-α, and so on may alter the prevalence of HT. Since there is such a risk (albeit small) of these disorders, it is more feasible that individuals receiving these medications undergo thyroid function tests and thyroid Abs [176,177,178].

It is difficult to accurately determine the directionality and magnitude of factors that may affect the overall prevalence of HT; e.g., the vast majority of the studies did not evaluate or stratify the target population according to these factors, nor did they evaluate the possible interactions between them (which clearly exist, including in the context of an AID) (Figure 2).

Finally, other less studied factors that may influence the prevalence of HT should also be taken into account, such as stress, alterations in the intestinal microbiome, and the presence of other infections (bacterial, fungal, or parasitic) [179,180]. However, although biologically plausible, in the sense that these factors are capable of stimulating an autoimmune response (and increasing the risk of HT), studies evaluating this aspect are scarce.

## 5. Strengths and Weaknesses of the Present Review

A notable strength of this scoping review is the exhaustive process used to collect data and select the studies reviewed. This involved an extensive search of bibliographic evidence in four major databases (MEDLINE, Web of Science, PubMed, and Scopus), allowing for a sensitive and robust search, offering a broad and accurate view of the topic. The geographical representation included studies from Europe, Africa, Asia, Oceania, North America, and South America, providing a broader view of the distribution of HT prevalence worldwide.

Likewise, several weaknesses are also evident, e.g., the language limitation regarding the choice of studies, the heterogeneity found, and the fact that there were several studies in which the diagnostic method used was not described; additionally, the evaluation (and diagnostic) criteria were also different between the studies. These considerations precluded a meta-analysis of the available information; however, we tried to mitigate these aspects by showing the results in a narrative and condensed manner according to variables such as country, continent, income group, study design, and the diagnostic method used.

### Future Implications

Large-scale, population-based studies should be conducted to analyze possible confounding factors and interactions between them in different geographic, social, economic, ethnic minority, and other settings. In light of current evidence, the results of studies that have evaluated the prevalence of HT should only be extrapolated to similar populations.

## 6. Conclusions

In summary, the global prevalence of HT is widely variable, estimated to be between 5% and 10%, with areas showing prevalences as high as >20% and as low as <0.5%. This prevalence is higher in women than in men. Multiple factors, such as diet, socioeconomics, diet, racial disparities and variations, and overdiagnosis, can affect the global prevalence of HT. These factors, combined with genetic/epigenetic predisposition and other environmental and lifestyle factors, form the conceptual framework of this disease.

## Figures and Tables

**Figure 1 medsci-13-00043-f001:**
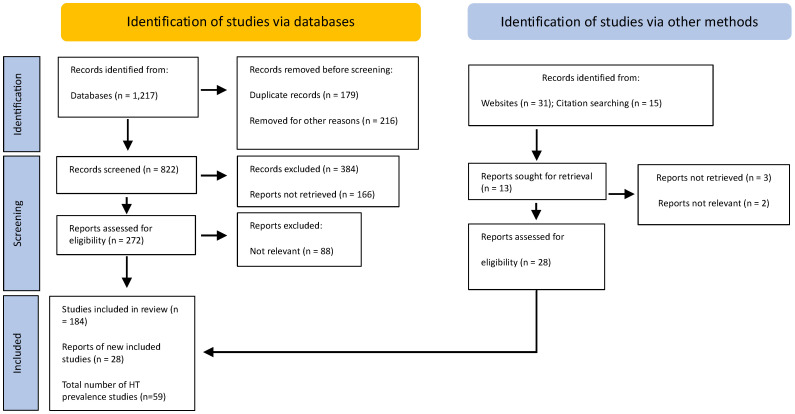
A PRISMA flow diagram. The method for the selection of articles.

**Figure 2 medsci-13-00043-f002:**
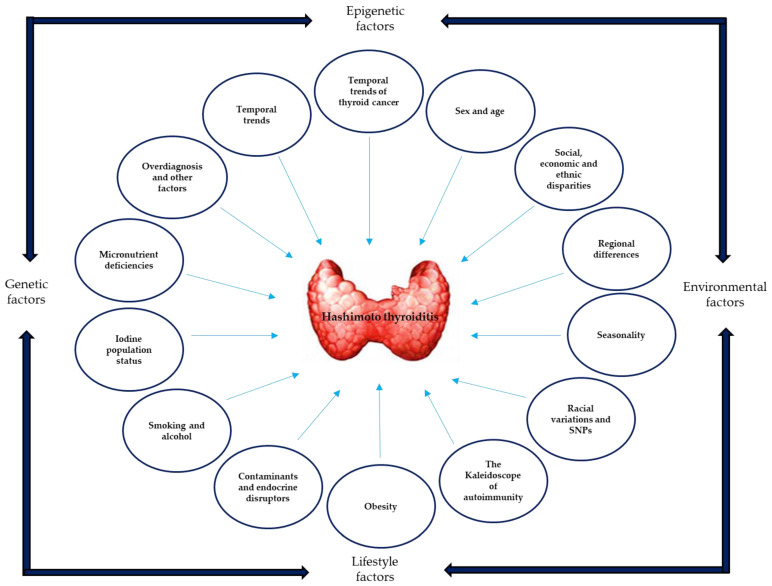
The conceptual framework of HT has a genetic/epigenetic basis, with an environmental and lifestyle component; the prevalence of HT is widely variable, probably influenced by multiple factors, which can interact with each other, substantially modifying the worldwide distribution of the disease.

**Table 1 medsci-13-00043-t001:** The global prevalence of Hashimoto’s thyroiditis in Europe (n = 25 studies), according to country, income group, study design, and sample type.

Author (Year) [Ref.]	Country (Income Group)	Case/Total	Study Design	Diagnostic Criteria	% Prevalence of Hashimoto’s Thyroiditis (95% CI)
Dingle PR, et al. (1966) [15]	England (HI)	52/469	Cross-sectional (PBS)	Serum (Abs)	11 (8.4–14.3)
Jacobs A, et al. (1969) [16]	England (HI)	99/989	Cross-sectional (PBS)	Serum (Abs)	10 (8.2–12.1)
Tunbridge WM, et al. (1977) [18]	England (HI)	56/2779	Cross-sectional	Serum (Abs)	2.0 (1.5–2.6)
Prentice LM, et al. (1990) [17]	England (HI)	124/698	Cross-sectional (PBS)	Serum (Abs)	17.8 (15–20.8)
Aho K, et al. (1971) [19]	Finland (HI)	89/1137	Cross-sectional (CBS)	Serum (Abs)	7.8 (6.3–9.5)
Gordin A, et al. (1972) [20]	Finland (HI)	282/2961	Cross-sectional (PBS)	Serum (Abs)	9.5 (8.5–10.6)
Bjøro T, et al. (1984) [21]	Norway (HI)	56/1643	Cross-sectional (PBS)	Serum (Abs)	3.4 (2.6–4.4)
Bryhni B, et al. (1996) [22]	Norway (HI)	176/2551	Cross-sectional (PBS)	Serum (Abs)	6.9 (5.9–8.0)
Völzke H, et al. (2003) [23]	Germany (HI)	47/3941	Cross-sectional	Serum (Abs)	1.2 (0.9–1.6)
Döbert N, et al. (2008) [24]	Germany (HI)	98/700	Cross-sectional (PBS)	Serum (Abs + TU)	14 (11.5–16.8)
Khattak RM, et al. (2016) [25]	Germany (HI)	358/4420	Cohort	Serum (Abs)	8.1 (7.3–8.9)
Aghini-Lombardi F, et al. (1999) [26]	Italy (HI)	50/1411	Cross-sectional (PBS)	Serum (Abs)	3.5 (2.6–4.6)
Benvenga S, et al. (2008) [27]	Italy (HI)	4064/23,000	Array research (CBS)	Serum, thyroid tissue (Abs + TU + FNA)	17.7 (17.2–18.2)
Sardu C, et al. (2012) [28]	Italy (HI)	678/25,885	Cross-sectional (PBS)	NR (NR)	2.6 (2.4–2.8)
Aghini–Lombardi F, et al. (2013) [29]	Italy (HI)	224/1065	Cross-sectional (PBS)	Thyroid tissue (Abs + TU)	21 (18.6–23.6)
Tammaro A, et al. (2016) [30]	Italy (HI)	2828/7976	Array research (PBS)	Serum (Abs)	35.5 (34.4–36.5)
Pilli T, et al. (2019) [31]	Italy (HI)	9/142	Cross-sectional	Serum (Abs)	6.3 (2.9–11.7)
Pedersen IB, et al. (2003) [32]	Denmark (HI)	787/4184	Cross-sectional (PBS)	Serum (Abs)	18.0 (17.6–20.0)
Pedersen IB, et al. (2011) [33]	Denmark (HI)	778/3570	Cross-sectional (PBS)	Serum (Abs)	21.8 (20.4–23.1)
Møllehave LT, et al. (2024) [34]	Denmark (HI)	979/2465	Cross-sectional (PBS)	Serum (Abs)	39.7 (37.8–41.6)
Valdés S, et al. (2017) [35]	Spain (HI)	391/4554	Cross-sectional (PBS)	Serum (Abs)	8.6 (7.8–9.4)
Józków P, et al. (2017) [36]	Poland (HI)	29,375/586,703	Cross-sectional (CBS)	Serum (Abs)	5.0 (5.0–5.1)
Troshina EA, et al. (2021) [37]	Russia (UMI)	428/100,000	Cross-sectional (PBS)	Serum (Abs)	0.42 (0.38–0.46)
Izic B, et al. (2021) [38]	Bosnia and Herzegovina (UMI)	358/82,000	Array research (PBS)	Serum (Abs)	0.43 (0.39–0.48)
Strikić Đula I, et al. (2022) [39]	Croatia (HI)	1044/4402	Cross-sectional (PBS)	Serum (Abs)	23.7 (22.4–25.9)

Abbreviations: Abs: antibodies; CBS: clinic-based study; HI: high income; NR: not reported; PBS: population -based study; TU: thyroid ultrasound; UMI: upper-middle income.

**Table 2 medsci-13-00043-t002:** The global prevalence of Hashimoto’s thyroiditis in Africa (n = 3 studies), according to country, income group, study design, and sample type.

Author (Year) [Ref.]	Country (Income Group)	Case/Total	Study Design	Diagnostic Criteria	% Prevalence of Hashimoto’s Thyroiditis (95% CI)
Chabchoub G, et al. (2006) [40]	Tunisia (LMI)	246/1079	Array research (CBS)	Serum (Abs)	22.8 (20.3–25.4)
Okosieme OE, et al. (2007) [41]	Nigeria (LMI)	7/104	Cross-sectional (CBS)	Serum (Abs)	6.7 (2.7–13.4)
Sarfo-Kantanka O, et al. (2017) [42]	Ghana (LMI)	583/8099	Retrospective cohort (HBS)	Serum (Abs + TU)	7.2 (6.6–7.8)

Abbreviations: Abs: antibodies; CBS: clinic-based study; HBS: hospital-based data; LMI: low-middle income; TU: thyroid ultrasound.

**Table 3 medsci-13-00043-t003:** The global prevalence of Hashimoto’s thyroiditis in Oceania (n= 2 studies), according to country, income group, study design, and sample type.

Author (Year) [Ref.]	Country (Income Group)	Case/Total	Study Design	Diagnostic Criteria	% Prevalence of Hashimoto’s Thyroiditis (95% CI)
O’Leary PC, et al. (2006) [43]	Australia (HI)	282/2115	Cross-sectional health survey	Serum (Abs)	13.3 (11.8–14.8)
Deshpande P, et al. (2016) [44]	Australia (HI)	17/198	Cross-sectional (PBS)	Serum (Abs)	8.6 (5.1–13.4)

Abbreviations: Abs: antibodies; HI: high income; PBS: population-based study.

**Table 4 medsci-13-00043-t004:** The global prevalence of Hashimoto’s thyroiditis in Asia (n = 19 studies), according to country, income group, study design, and sample type.

Author (Year) [Ref.]	Country (Income Group)	Case/Total	Study Design	Diagnostic Criteria	% Prevalence of Hashimoto’s Thyroiditis (95% CI)
Okayasu I, et al. (1991) [45]	Japan (HI)	328/1826	Cross-sectional	Thyroid tissue	18 (0.16–0.19)
Konno N, et al. (1993) [46]	Japan (HI)	457/4110	Cross-sectional (PBS)	Serum (Abs)	11.1 (10.2–12.1)
Morinaka S, et al. (1995) [47]	Japan (HI)	61/6348	Cross-sectional (CBS)	Serum (Abs) + TU + FNA)	1.0 (0.7–1.2)
Nagata K, et al. (1998) [48]	Japan (HI)	142/1039	Cross-sectional (PBS)	Serum (Abs)	13.7 (11.6–15.9)
Kurata S, et al. (2007) [49]	Japan (HI)	25/1626	Cross-sectional (CBS)	Serum, thyroid tissue (Abs + TU + FNA)	1.5 (1.0–2.3)
Teng W, et al. (2006) [50]	China (UMI)	32/3761	Cross-sectional study	Serum (Abs)	0.9 (0.6–1.2)
Teng X, et al. (2008) [51]	China (UMI)	67/778	Array research (PBS)	Serum (Abs + TU)	8.6 (6.7–10.8)
Li Y, et al. (2008) [52]	China (UMI)	353/3018	Cross-sectional (PBS)	Serum (Abs) + TU)	11.7 (10.6–12.9)
Teng X, et al. (2011) [53]	China (UMI)	363/3813	Cross-sectional	Serum (Abs) + TU	9.5 (8.6–10.5)
Wu Q, et al. (2015) [54]	China (UMI)	172/6152	Cross-sectional (PBS)	Serum (Abs)	2.8 (2.4–3.2)
Gu F, et al. (2016) [55]	China (UMI)	17/5293	Cross-sectional (PBS)	Serum (Abs) + TU)	0.3 (0.2–0.5)
Li Y, et al. (2016) [56]	China (UMI)	187/2856	Array research	Serum (Abs)	6.5 (5.7–7.5)
Wan S, et al. (2020) [57]	China (UMI)	198/1225	Cross-sectional survey	Serum (Abs)	16.1 (14.1–18.2)
Chen Y, et al. (2021) [58]	China (UMI)	298/2946	Cross-sectional (PBS)	Serum (Abs) + TU)	10.1 (9.0–11.3)
Yu Z, et al. (2021) [59]	China (UMI)	148/1159	Cross-sectional (PBS)	Thyroid tissue	12.8 (10.9–14.8)
Fernando RF, et al. (2012) [60]	Sri Lanka (LMI)	353/5200	Cross-sectional (PBS)	Serum (Abs)	6.8 (6.1–7.5)
Amouzegar A, et al. (2017) [61]	Iran (LMI)	742/5783	Cross-sectional (PBS)	Serum (Abs)	12.8 (12–13.7)
Kim HJ, et al. (2021) [62]	South Korea (HI)	29,429/21,705,883	Array research (PBS)	NR (NR)	0.1 (0.1–0.1)
Ajlouni KM, et al. (2022) [63]	Jordan (UMI)	567/3753	Cross-sectional (PBS)	Serum (Abs)	15.1 (13.9–16.2)

Abbreviations: Abs: antibodies; CBS: clinic-based study; FNA: fine needle aspiration; HI: high income; LMI: low-middle income; NR: not reported; PBS: population-based study; TU: thyroid ultrasound; UMI: upper-middle income.

**Table 5 medsci-13-00043-t005:** The global prevalence of Hashimoto’s thyroiditis in North America (four studies), according to country, income group, study design, and sample type.

Author (Year) [Ref.]	Country (Income Group)	Case/Total	Study Design	Diagnostic Criteria	% Prevalence of Hashimoto’s Thyroiditis (95% CI)
Okayasu I, et al. (1994) [64]	USA (HI)	457/2040	Cross-sectional (PBS)	Thyroid tissue (pathological section)	22.4 (20.6–24.3)
Flores-Rebollar A, et al. (2015) [65]	Mexico (UMI)	36/427	Cross-sectional (PBS)	Serum (Abs) + TU)	8.4 (6.0–11.5)
Caturegli G, et al. (2016) [66]	USA (HI)	4/1075	Cross-sectional (PBS)	NR (NR)	0.4 (0.1–0.91)
Zhang X, et al. (2024) [67]	USA (HI)	4454/33,117	Cross-sectional study	Serum (Abs)	13.4 (13.1–13.8)

Abbreviations: Abs: antibodies; HI: high income; NR: not reported; PBS: population-based study; TU: thyroid ultrasound; UMI: upper-middle income.

**Table 6 medsci-13-00043-t006:** The global prevalence of Hashimoto’s thyroiditis in South America (n = 6 studies), according to country, income group, study design, and sample type.

Author (Year) [Ref.]	Country (Income Group)	Case/Total	Study Design	Diagnostic Criteria	% Prevalence of Hashimoto’s Thyroiditis (95% CI)
Tomimori E, et al. (1995) [68]	Brazil (UMI)	72/547	Cross-sectional (PBS)	TU	13.2 (10.4–16.3)
Camargo RY, et al. (2006) [69]	Brazil (UMI)	82/420	Cross-sectional	Serum (Abs) + TU	19.5 (15.8–23.6)
Camargo RY, et al. (2008) [70]	Brazil (UMI)	183/1085	Cross-sectional (PBS)	Serum (Abs)	16.9 (14.7–19.2)
Vecchiatti SM, et al. (2015) [71]	Brazil (UMI)	106/4613	Cross-sectional (CBS)	Thyroid tissue	2.3 (1.9–2.8)
Tolentino Júnior DS, et al. (2019) [72]	Brazil (UMI)	85/60,413	Cross-sectional	NR	0.1 (0.1–0.2)
Vargas-Uricoechea, H, et al. (2023) [73]	Colombia (UMI)	2150/9638	Cross-sectional (PBS)	Serum (Abs)	22.3 (20.6–24.0)

Abbreviations: Abs: antibodies; CBS: clinic-based study; NR: not reported; PBS: population-based study; TU: thyroid ultrasound; UMI: upper-middle income.

## Data Availability

The data presented in this study are available on request from the corresponding author.

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
