# Peer review of "A Scoping Review on the Prevalence of Hashimoto’s Thyroiditis and the Possible Associated Factors"

_medsci, 2025, doi:10.3390/medsci13020043_

Round 1
Reviewer 1 Report
Comments and Suggestions for Authors
Review of the manuscript titled "A Scoping Review on Prevalence of Hashimoto's Thyroiditis and Possible Associated Factors"
1.Scientific Quality and Originality
The manuscript addresses an important and clinically relevant topic — the global prevalence of Hashimoto’s thyroiditis and its associated factors. This topic is timely due to the growing importance of autoimmune thyroid diseases and their complex pathogenesis.
Strengths:
Extensive literature search across multiple databases.
Inclusion of epidemiological data from around the world.
In-depth discussion of genetic, epigenetic, environmental, and lifestyle factors.
Limitations:
No quantitative (meta-analytical) analysis was performed, although the authors justify this due to the heterogeneity of the data.
2. Clarity and Structure
The article is well-organized, logically structured, and clearly written.
3. References and Literature Review
The cited sources are up-to-date and appropriate. The authors effectively place their findings within the context of existing scientific knowledge.
4. Methodological Rigor
The scoping review was conducted in accordance with PRISMA and JBI guidelines. The inclusion criteria and search strategy are clearly described.
5. Significance and Impact
The article provides a valuable contribution to the understanding of global variability in HT and highlights the need to standardize diagnostic criteria. It may be of practical value to clinicians, researchers, and public health professionals.
Author Response
Answer to Reviewer 1:
We are infinitely grateful to the reviewer for the way he or she describes our manuscript; his or her words have enriched us greatly. Based on the concepts outlined by the reviewer, we have no responses or changes to make to the article.
Cordially,
Hernando Vargas-Uricoechea.
First author and corresponding author.

Reviewer 2 Report
Comments and Suggestions for Authors
The article entitled “A Scoping Review on Prevalence of Hashimoto's Thyroiditis and Possible Associated Factors” is review summarising literature knowledge. The authors are determining the global prevalence of Hashimoto's thyroiditis and analyse the possible factors that may influence the population variability of the disease. Overall, it is a very well written and extremely interesting manuscript. The authors have to address some (very) minor points before this study can be published in the journal “medical sciences”.
My minor concerns are:
- Sometimes citations for statements are missing; the authors have to prove all statements with citations. It is highly recommended to add citations to lines 52, 55 and to chapter 3.1
- In all six table “et al.” must be added to the name of the first author
Author Response
The article entitled “A Scoping Review on Prevalence of Hashimoto's Thyroiditis and Possible Associated Factors” is review summarising literature knowledge. The authors are determining the global prevalence of Hashimoto's thyroiditis and analyse the possible factors that may influence the population variability of the disease. Overall, it is a very well written and extremely interesting manuscript. The authors have to address some (very) minor points before this study can be published in the journal “medical sciences”.
My minor concerns are:
- Sometimes citations for statements are missing; the authors have to prove all statements with citations. It is highly recommended to add citations to lines 52, 55 and to chapter 3.1
Answer to reviewer 2:
Many thanks to the reviewer for their comments and suggestions. We fully agree with the changes to be made and have of course made them: lines 52, 55 and to chapter 3.1 (The changes made have been highlighted in yellow).
- In all six table “et al.” must be added to the name of the first author.
Answer to reviewer 2:
Many thanks to the reviewer for their comment and suggestion.
We fully agree, so we have inserted the term: et al. in each of the manuscript's tables.
We also made some minor changes to Figure 1 (adjusting the spacing and font size, as they were not fully visible).
Additionally, we made a couple of changes to the table titles (we believe the autocorrector used the term “prevalencia” when the correct term is prevalence), and we apologize for that.
Each of these changes were highlighted in yellow.
Again, many thanks to reviewer 2, for all his help, collaboration, insights, and suggestions.
Cordially,
Hernando Vargas-Uricoechea.
First author and corresponding author.
